# Integrating Multi-Organ Imaging-Derived Phenotypes and Genomic Information for Predicting the Occurrence of Common Diseases

**DOI:** 10.3390/bioengineering11090872

**Published:** 2024-08-28

**Authors:** Meng Liu, Yan Li, Longyu Sun, Mengting Sun, Xumei Hu, Qing Li, Mengyao Yu, Chengyan Wang, Xinping Ren, Jinlian Ma

**Affiliations:** 1Human Phenome Institute, Fudan University, Shanghai 201203, China; 21210880006@m.fudan.edu.cn (M.L.); 22212030011@m.fudan.edu.cn (L.S.); 23212030012@m.fudan.edu.cn (M.S.); 13188816530@163.com (Q.L.); yumengyao@fudan.edu.cn (M.Y.); wangcy@fudan.edu.cn (C.W.); 2Department of Radiology, Ruijin Hospital, School of Medicine, Shanghai Jiao Tong University, Shanghai 200025, China; ly40730@rjh.com.cn; 3Ultrasound Department, Ruijin Hospital, School of Medicine, Shanghai Jiao Tong University, Shanghai 200025, China; 4Radiology Department, Jiangyin Affiliated Hospital of Nanjing University of Chinese Medicine, 130 Renmin Middle Road, Jiangyin 214400, China

**Keywords:** disease prediction, medical imaging, imaging-derived phenotypes, genomic information, polygenic risk scores

## Abstract

As medical imaging technologies advance, these tools are playing a more and more important role in assisting clinical disease diagnosis. The fusion of biomedical imaging and multi-modal information is profound, as it significantly enhances diagnostic precision and comprehensiveness. Integrating multi-organ imaging with genomic information can significantly enhance the accuracy of disease prediction because many diseases involve both environmental and genetic determinants. In the present study, we focused on the fusion of imaging-derived phenotypes (IDPs) and polygenic risk score (PRS) of diseases from different organs including the brain, heart, lung, liver, spleen, pancreas, and kidney for the prediction of the occurrence of nine common diseases, namely atrial fibrillation, heart failure (HF), hypertension, myocardial infarction, asthma, type 2 diabetes, chronic kidney disease, coronary artery disease (CAD), and chronic obstructive pulmonary disease, in the UK Biobank (UKBB) dataset. For each disease, three prediction models were developed utilizing imaging features, genomic data, and a fusion of both, respectively, and their performances were compared. The results indicated that for seven diseases, the model integrating both imaging and genomic data achieved superior predictive performance compared to models that used only imaging features or only genomic data. For instance, the Area Under Curve (AUC) of HF risk prediction was increased from 0.68 ± 0.15 to 0.79 ± 0.12, and the AUC of CAD diagnosis was increased from 0.76 ± 0.05 to 0.81 ± 0.06.

## 1. Introduction

Advanced application technologies based on biomedical imaging have the potential to significantly enhance diagnostic efficiency and accuracy. The fusion of biomedical imaging and genomics information for disease classification is a cutting-edge approach in medical diagnostics [1]. This technique facilitates a more comprehensive integration of genetic and environmental factors contributing to complex diseases, such as cardiovascular disease, diabetes, and liver disease, which are now increasingly studied using multi-omics approaches [2,3,4,5]. Medical imaging provides visual information about the anatomy and functional status within the body [6]. Phenotypes derived from imaging can quantitatively reflect the structure and functional status of organs, serving as excellent biomarkers for disease prediction [7]. On the other hand, polygenic risk scores (PRSs) are widely employed in research due to their demonstrated validity and potential clinical utility in predicting various common diseases [8]. They provide crucial insights into inherent genetic factors that influence disease, aiding significantly in early disease diagnosis [9]. They offer valuable insights into disease progression and facilitate early detection, independent of disease presence or activity levels [10]. Integrating cardiac magnetic resonance (CMR) traits and PRSs can enhance the performance of predicting complex traits and diseases [5]. Integrating PRSs with the QCancer-10 score [11], which is calculated relatively easily from health records, modestly improves risk prediction over the use of the Qcancer-10 score alone [12]. Wang et al. developed a classifier incorporating both MRI and PRS features, which achieved the optimal prediction performance in schizophrenia [13]. However, a significant limitation of previous studies is their reliance on IDPs from a single organ, overlooking the potential benefits of integrating data from multiple organs. This limitation may constrain our understanding of disease mechanisms and reduce the comprehensiveness of treatment strategies. As healthcare systems increasingly embrace genomic medicine, there are significant opportunities to integrate PRSs, which summarize an individual’s genetic predisposition for adverse treatment outcomes and disease complications, together with imaging-derived phenotypes (IDPs) into clinical decision-making. 

In the present study, we analyzed IDPs from various organs and disease-related PRSs of disease from the UK Biobank (UKBB) [14] dataset to construct a disease prediction model. UKBB represents a multi-center dataset, with inpatient record data sourced from HES (England), PEDW (Wales), or SMR (Scotland). Additionally, we evaluated the performance enhancements of fusion models, which integrate both IDPs and PRSs, compared to models utilizing solely IDPs or PRSs. The diseases considered in our analysis were common ones that affect multiple organs. The process of constructing the disease prediction model is depicted in the Figure 1.

## 2. Materials and Methods

### 2.1. Study Design

The UKBB is a prospective cohort of approximately 500,000 individuals from the United Kingdom, enrolled between 2006 and 2010. The cohort includes extensive phenotyping, imaging, and multiple genomic data types. The design of the cohort has been detailed in a previous paper [14]. Starting in 2014, about 40,000 participants returned for their first multi-modal imaging visit, which included brain MRI, heart MRI, and abdominal MRI. Representative multi-modal images are illustrated in Figure 2. This comprehensive imaging allows for the evaluation of organ conditions across the entire body. Longitudinal health outcomes for the study participants are tracked via national health datasets. Data from the UKBB (Application Number: 96511) were applied for the present analysis.

### 2.2. Baseline Examination and Sample Collection

Our study focused on the PRSs of diseases and IDPs from multiple organs, all accessible within the UKBB dataset. The participant selection process in our study is illustrated in Figure 3, which details the inclusion and exclusion criteria used to define our study cohorts from the UKBB population. We focused on participants of European descent because they constitute the majority in the UKBB. Additionally, performing PRS calculations requires a homogeneous population to ensure the accuracy and reliability of the genetic associations, minimizing potential biases due to population stratification. Ultimately, 8646 individuals remained with complete IDPs and PRSs, and their baseline characteristics at the time of their first visit are summarized in Table 1 below.

In our study, we used specific IDPs from different organs including the heart, brain, kidney, liver, lung, pancreas, and spleen from the UKBB, with the corresponding UKBB Field IDs detailed in Table 2. Those IDPs were automatically extracted through the deep learning model, and detailed information about the process of IDP extraction can be found in the reference. Cardiac imaging data included the phenotypes from cardiac and aortic structure and function. For the kidney, Langner et al. [15] automatically segmented the renal parenchyma and extracted IDPs related to the renal parenchyma volume from abdominal MRI scans. Zhao et al. [16] extracted brain phenotypes from T1-weighted images. Liver phenotypes were provided by Mojtahed et al. [17], using proton density fat fraction images (Field ID 40061), gradient echo images (Field ID 20203), and IDEAL sequence images (Field ID 20254), which facilitated the calculation of liver fat fractions and corrected T1 (cT1) measurements. The volumes, fat fractions, and iron contents of the liver, spleen, and pancreas were extracted from abdominal MRI scans by Liu et al. [18]. Lastly, lung function phenotypes were assessed using spirometry tests, which measured multiple functional parameters including forced vital capacity (FVC), forced expiratory volume in one second (FEV1), peak expiratory flow (PEF), and the FEV1/FVC ratio. These IDPs serve as indicators of the structural and functional condition of the respective organs and play a crucial role in our comprehensive analysis of the associations between organ health and common diseases.

In this study, nine common diseases were included for analysis, namely atrial fibrillation (AF), heart failure (HF), hypertension, myocardial infarction (MI), asthma, type 2 diabetes (T2D), chronic kidney disease (CKD), coronary artery disease (CAD), and chronic obstructive pulmonary disease (COPD), classified according to the International Classification of Diseases (ICD-10) codes. And patients with multiple diseases were included, allowing for the possibility of participants being diagnosed with more than one disease simultaneously. Health was defined relative to the included diseases. Participants were classified as belonging to the healthy group if they did not have any of the specified diseases mentioned above, resulting in 5669 individuals being categorized in the healthy group. Specific ICD-10 codes and numbers of cases related to these diseases are listed in Table 3.

### 2.3. PRS Calculation

In this study, we calculated PRSs for nine common diseases. The computation procedure, detailed below, aimed to identify single-nucleotide polymorphisms (SNPs) significantly associated with these diseases while excluding any SNPs in linkage disequilibrium (LD).

Initially, summary statistics for each disease were processed using PLINK [20] software version 1.07. This process utilized a *p*-value-based clumping method with specific parameters: --clump-p1 = 0.0001, --clump-p2 = 0.01, --clump-r2 = 0.5, and --clump-kb = 250. These settings were selected to identify SNPs strongly associated with the diseases, yet independent of LD effects. European population data from the 1000 Genomes Project [21] served as the reference panel for LD. Subsequent steps involved the extraction and merging of the selected SNP data across different chromosomes, facilitated by bgenix [22] and cat-bgen software tools for handling the original UKBB per-chromosome genetic data files. The dataset was further processed by converting the bgen files to PLINK format, with the --hard-call-threshold parameter set to 0.1.

Quality control for the SNPs was conducted by leveraging imputed genotype quality (INFO) and minor allele frequency (MAF) data provided by the UKBB, utilizing QCTOOL [23] for the calculations. This crucial step involved removing ambiguous SNPs and those with INFO values below 0.4 or MAF values less than 0.005. The analysis specifically targeted samples of samples of European ancestry from the UKBB. Subsequently, PRSs were calculated using the quality-controlled data. This rigorous process guarantees the generation of reliable PRSs, offering valuable insights into the genetic predisposition towards common diseases in the population.

### 2.4. Prediction Model

The study employed multi-organ IDPs and PRSs to predict disease outcomes. Our analysis compared the performance of three distinct models: the PRSs-Only Model, which used solely PRSs derived from genetic data; the IDPs-Only Model, which utilized only the IDPs from multiple organs, including quantitative measurements and features reflecting the structure and function of various organs; and the Combined PRSs and IDPs Model, which integrated both PRSs and multi-organ IDPs to enhance predictive power. Logistic regression with L1 norm regularization, implemented via the glmnet package in R (version 4.3.1), served as the prediction model. To address the imbalance between positive and negative samples in our dataset, a downsampling method was employed through the utilization of the R package themis. The data were split into a training set and a test set with a 7:3 ratio. Our analysis compared the performance of different models: using only PRSs, only IDPs, and a combination of both.

To explore the influence of lifestyle factors on disease prediction, we included the characteristics of smoking and drinking. The disease prediction accuracy was compared with and without lifestyle factors with the fusion model of IDPs and PRSs as the baseline. The process of model development was consistent with the methods previously described.

### 2.5. Validation Cohort

To validate our results observed in the entire UKBB dataset, the models were tested in HES, the largest center within the UKBB. Further details on the model and analysis in this center remain as those previously described.

### 2.6. Feature Ranking

Feature importance was determined by fitting a random forest model, using participants’ variables as inputs and the predictions of our model as outputs. To evaluate feature importance, the mean decrease in Gini coefficient, a metric derived from the random forest algorithm, was utilized. This approach permitted the assessment the contribution of each feature within our IDPs + PRSs model and generated a ranked order of feature importance for the various IDPs and PRSs. This ranking is essential for understanding how IDPs and PRSs differentially impact the predictive accuracy of our disease prediction models.

## 3. Results

### 3.1. Prediction Results

To evaluate the performance of disease prediction models, our study focused on three distinct methodologies: models based only on PRSs, models based only on multi-organ IDPs, and models integrating both PRSs and IDPs. The effectiveness of these models was evaluated by AUC, a robust metric for evaluating the predictive accuracy of binary classifiers. The integrated model, combining both PRSs and IDPs, demonstrated significant improvements in predictive performance compared to models using either PRSs or IDPs alone in T2D, AF, CAD, COPD, asthma, MI, and HF. For instance, the combined model of CAD demonstrated superior predictive accuracy with an AUC of 0.81 ± 0.06, compared to 0.76 ± 0.05 for IDPs alone and 0.66 ± 0.06 for PRSs alone. Similarly, the PRS + IDP model (AUC = 0.79 ± 0.12) of HF was superior to both PRS (AUC = 0.63 ± 0.16) and IDP models (AUC = 0.68 ± 0.15). However, in other diseases such as hypertension, the combined model did not perform as well as the other models. The IDP-alone model achieved an AUC of 0.77 ± 0.03, which was more effective than the combined approach (AUC = 0.73 ± 0.03). The detailed ROCs for the common disease prediction models are shown in Figure 4, and the performance metrics for our disease prediction models, including the Pearson correlation between the true and predicted conditions of the diseases, AUC, sensitivity, specificity, and accuracy, are summarized in Table 4. The significance of the differences between the AUCs was calculated using https://www.medcalc.org/calc/comparison_of_independentROCtest.php accessed on 16 July 2024.

Table 5 summarizes the prediction performance for various diseases with and without the inclusion of lifestyle factors. The data indicated that integrating lifestyle factors into the prediction model generally decreases the predictive performance of most diseases. Only CKD, asthma, and CAD demonstrated improved performance, although the enhancements were not statistically significant.

### 3.2. Validation

A subset of UKBB data facilitated the evaluation of the robustness and generalizability of our models. The AUC results were compared between the entire UKBB dataset and the HES center (Figure 5). In the entire UKBB dataset, the integrated model, combining both PRSs and IDPs, demonstrated significant improvements in predictive performance compared to models using either PRSs or IDPs alone in T2D, AF, CAD, COPD, asthma, MI, and HF. In the HES center, the integrated model performed the best in CKD, COPD, AF, HF, hypertension, MI, T2D.

### 3.3. Feature Ranking

Figure 6 illustrates the feature importance derived from our models, showcasing the differential contributions of PRSs and multi-organ IDPs across various disease predictions. For AF, left-atrium-related features emerged as the most crucial. For T2D, features related to the liver were important. For hypertension prediction, features related to the thickness of the myocardium and the distensibility of Ao made a great contribution to the prediction model. For CAD and COPD prediction, PRSs contributed more than other IDPs. A change in kidney parenchyma volume is a potential marker for the presence of CKD. Lung function parameter was a major contributor in predictive models for both COPD and asthma, emphasizing its recognized impact. For HF prediction, the ejection fraction of the ventricle played the most important role among all the features. Surprisingly, for MI prediction, the fat fraction of pancreas and liver iron corrected T1 value contributed more than cardiac IDPs.

## 4. Discussion

In recent years, the integration of multi-modal data for disease prediction has become increasingly prevalent, reflecting a broader trend in precision medicine of harnessing diverse datasets for enhanced diagnostic accuracy. For instance, Dolci et al. [24] introduced a deep multi-modal generative data fusion framework for integrating neuroimaging and genomics in classifying Alzheimer’s disease, achieving superior prediction performance despite incomplete data availability. Vanguri et al. [25] demonstrated the enhanced predictive capacity of integrating CT imaging and genomic features to forecast immunotherapy response in advanced non-small-cell lung cancer, achieving superior performance compared to individual biomarkers like tumor mutational burden and PD-L1 expression. However, previous research has predominantly concentrated on a limited number of diseases and single-organ imaging. Our research stands out by incorporating a wide array of diseases and a rich variety of IDPs from multiple organs, enhancing both the predictive power and clinical relevance of our models.

Our investigation focused on the fusion of genomic data with multi-organ IDPs to predict prevalent diseases, utilizing the extensive UKBB dataset. This integration yielded substantial enhancements in predictive accuracy, particularly evidenced by notable increases in the AUC for asthma, COPD, AF, CAD HF, MI, and T2D. However, not all diseases showed improved prediction. CKD and hypertension did not exhibit notable enhancements in predictive accuracy. This could be attributed to the relatively minor genetic contributions to these conditions, which may diminish the effectiveness of PRSs in their prediction. This insight underscored the importance of prioritizing the consideration of acquired dietary and lifestyle habits in the treatment of CKD and hypertension.

Our result of feature importance revealed organ-specific contributions to predictive accuracy. Specific features such as those related to the left atrium (LA) were pivotal for AF prediction, while liver-related features significantly influenced the prediction of T2D. These insights not only demonstrate the critical role of targeted organ analysis in enhancing disease prediction models but also suggest that individual organ metrics can be decisive factors in managing and understanding complex diseases. Recognizing these patterns enables a more refined approach to precision medicine, facilitating tailored treatment strategies based on specific organ involvement and genetic profiles.

Our study has several limitations. First, while we included a range of IDPs from the UKBB that have been previously extracted, several potentially valuable features were not considered. Notably, advanced imaging metrics such as cardiac T1 mapping [26] or specific modalities in brain imaging [27], such as functional MRI (fMRI) reflecting brain activity or diffusion MRI (dMRI) reflecting local tissue microstructure, were not included in our analysis. However, incorporating these techniques could potentially provide valuable insights into certain diseases. The inclusion of these additional features in future studies could significantly enhance both the predictive accuracy and clinical relevance of the models. Second, our study only included imaging and genetic data. Integrating additional data types like proteomics [28] and transcriptomics [29], which capture micro-level changes in the body, could enhance predictive models further [30,31,32]. These omics data offer insights into molecular pathways and functional alterations, potentially improving predictive accuracy. Last, logistic regression was selected for building a prediction model because of its interpretability, computational efficiency, and suitability for the data structure. Future research could explore more complex machine learning models to possibly improve outcomes, given a substantial dataset size and sufficient training time.

## Figures and Tables

**Figure 1 bioengineering-11-00872-f001:**
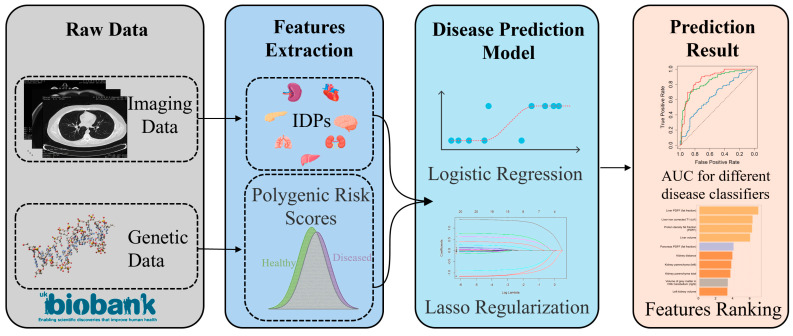
Workflow of disease prediction model development. This diagram illustrates the systematic process used in our study, beginning with the collection of raw data from the UK Biobank, including imaging and genetic data. The process involves the extraction of key features, specifically imaging-derived phenotypes (IDPs) and polygenic risk scores (PRSs), which are subsequently employed to develop disease prediction models using logistic regression model with lasso regularization. The performance of these models is evaluated by AUC for different disease classifiers and by the ranking of features based on their importance in prediction.

**Figure 2 bioengineering-11-00872-f002:**
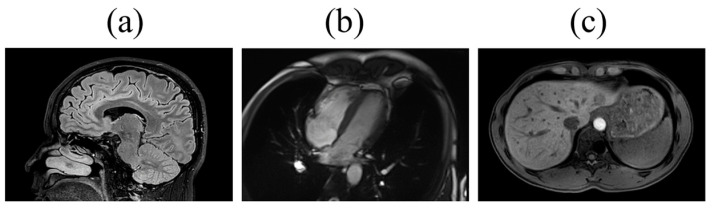
Multi-modal medical imaging. (**a**) Brain MRI; (**b**) heart MRI; (**c**) abdominal MRI.

**Figure 3 bioengineering-11-00872-f003:**
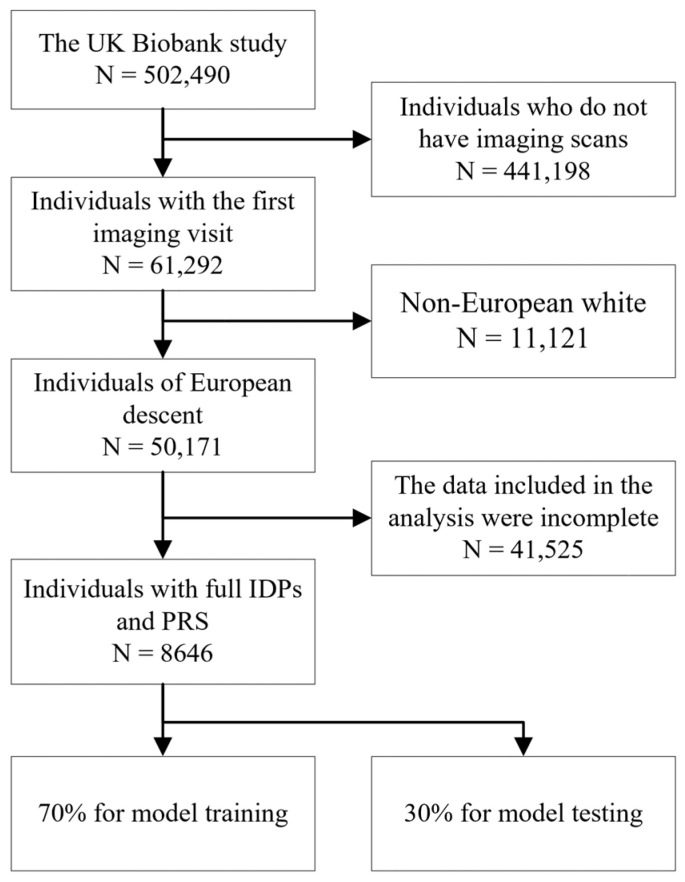
Inclusion and exclusion flowchart for the study.

**Figure 4 bioengineering-11-00872-f004:**
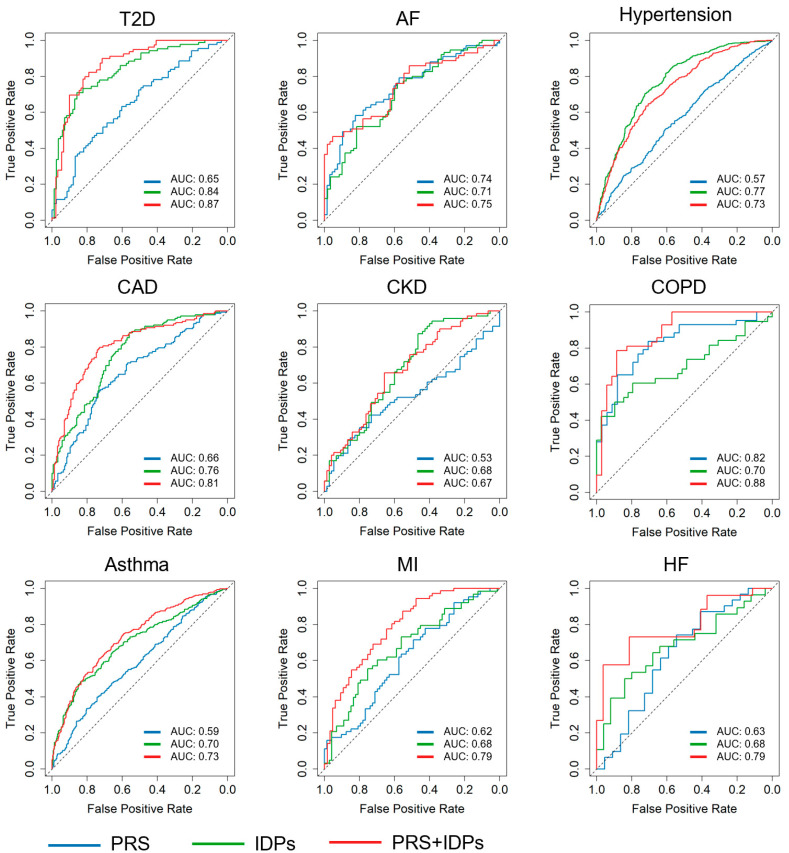
ROCs for common disease prediction models. This figure displays the ROCs for models predicting several common diseases, comparing the performance of models based on PRSs (blue line), IDPs (green line), and a combination of both PRSs and IDPs (red line). Each panel represents a different disease, with AUC values indicating the predictive accuracy of each model. CKD: chronic kidney disease; COPD: chronic obstructive pulmonary disease; AF: atrial fibrillation; CAD: coronary artery disease; HF: heart failure; MI: myocardial infarction; T2D: type 2 diabetes.

**Figure 5 bioengineering-11-00872-f005:**
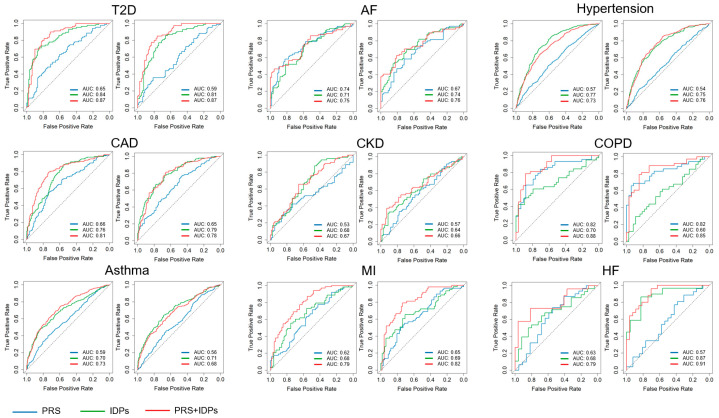
Comparative ROC analysis of disease prediction models across entire UKBB dataset and HES center. This figure presents the ROC curves for models predicting several common diseases, with panels for each disease split into two comparisons. The left panels utilize data from the whole UKBB, while the right panels are derived from the HES center. Each curve represents the performance of models based on PRSs in blue, IDPs in green, and a combination of both PRSs and IDPs in red. CKD: chronic kidney disease; COPD: chronic obstructive pulmonary disease; AF: atrial fibrillation; CAD: coronary artery disease; HF: heart failure; MI: myocardial infarction; T2D: type 2 diabetes.

**Figure 6 bioengineering-11-00872-f006:**
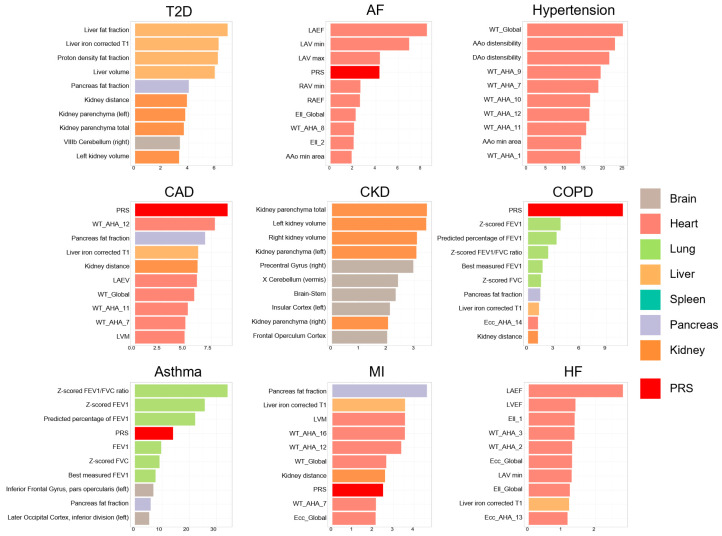
Feature importance in prediction models. This figure highlights the top 10 features for each disease prediction model. The color represents the specific organ from which these features are derived. The horizontal axis measures the features’ importance using the mean decrease in Gini, which quantifies their contribution to the predictive accuracy of the models.

**Table 1 bioengineering-11-00872-t001:** Population characteristics at the time of first visit with full IDPs and PRSs.

Characteristics	All (N = 8646)	Men (N = 3808)	Women (N = 4838)
Age (years)	64.2 ± 7.29	64.8 ± 7.4	63.7 ± 7.2
Weight (kg)	74.59 ± 13.82	82.54 ± 12.11	68.33 ± 11.71
Height (cm)	168.69 ± 8.99	175.98 ± 6.43	162.95 ± 6.09
BMI	26.25 ± 3.83	26.87 ± 3.39	25.77 ± 4.07
Leukocytes (×10^9^/L)	6.53 ± 1.90	6.53 ± 2.23	6.52 ± 1.59
RBC (×10^9^/L)	4.50 ± 0.40	4.75 ± 0.34	4.29 ± 0.32
Hemoglobin (g/L)	14.14 ± 1.22	15.02 ± 0.93	13.44 ± 0.95
Blood platelets (10^9^/L)	251.21 ± 56.30	236.74 ± 51.22	262.64 ± 57.49
ALT (U/L)	22.46 ± 13.02	26.68 ± 13.09	19.18 ± 11.99
AST (U/L)	25.26 ± 7.98	27.41 ± 8.09	23.59 ± 7.48
Dbil (μmol/L)	1.84 ± 0.80	2.01 ± 0.86	1.69 ± 0.70
Urea (mmol/L)	5.26 ± 1.21	5.50 ± 1.23	5.08 ± 1.17
C-reactive protein (mg/L)	1.84 ± 3.08	1.81 ± 3.08	1.86 ± 3.07
GGT (U/L)	32.51 ± 34.82	40.61 ± 35.87	26.19 ± 32.62
Lipoprotein (mg/L)	43.25 ± 48.87	44.47 ± 49.71	42.30 ± 48.20
TBIL (μmol/L)	9.37 ± 4.56	10.54 ± 5.01	8.45 ± 3.39
TGs (mmol/L)	1.61 ± 0.91	1.89 ± 1.04	1.39 ± 0.71

RBC: red blood cell count; ALT: alanine aminotransferase; AST: aspartate aminotransferase, Dbil: direct bilirubin, GGT: gamma glutamyltransferase, TBIL: total bilirubin, TGs: triglycerides, CKD: chronic kidney disease, COPD: chronic obstructive pulmonary disease, AF: atrial fibrillation, CAD: coronary artery disease, HF: heart failure, MI: myocardial infarction, T2D: type 2 diabetes.

**Table 2 bioengineering-11-00872-t002:** The Field IDs of the different organs’ IDPs.

Organ	IDPs	Field ID	Reference
Brain	volume of grey matter	25782–25920, 24360–24409	[16]
Heart	cardiac and aortic structure and function	24100–24181	[19]
Lung	$\mathrm{FVC}/\mathrm{FEV}1/\mathrm{PEF}/\frac{FEV1}{FVC}$/volume	3062, 3063, 3064, 20150, 20151, 20153, 20154, 20256, 20257, 20258, 21084	[18]
Liver	volume/fat fraction/iron/ corrected T1	21080, 21088, 21089, 40060, 40061, 40062	[17,18]
Spleen	volume/fat fraction/iron	21083, 21170, 21173	[18]
Pancreas	volume/fat fraction/iron	21087, 21090, 21091	[18]
Kidney	volume/kidney distance	21081, 21082, 21160–21163	[15]

FVC: forced vital capacity; FEV1: forced expiratory volume in one second; PEF: peak expiratory flow.

**Table 3 bioengineering-11-00872-t003:** The ICD10 code and number of cases of diseases.

Disease	ICD10 Code and Field ID	Number
HF	I110, I130, I132, Z941, T862, I500, I501, I509	89
MI	I252, I210, I211, I212, I213, I214, I219, I21X, I220, I221	226
AF	I480, I481, I482, I483, I484, I489	225
CAD	Z955, I252, Z951, I240, I241, I248, I249, I250, I251, I253, I254, I255, I256, I258, I259, I210, I211, I212, I213, I214, I219, I21X, I220, I221, I228, I229, I230, I231, I232, I233, I234, I235, I236, I238	497
T2D	E110, E111, E112, E113, E114, E115, E116, E117, E118, E119	284
Hypertension	I10, I110, I119, I120, I129, I130, I131, I132, I139, I150, I151, I152, I158, I159, O100, O101, O102, O103, O104, O109	1637
COPD	42016	128
Asthma	42014	1093
CKD	132032	244

HF: heart failure; MI: myocardial infarction; CKD: chronic kidney disease; COPD: chronic obstructive pulmonary disease; AF: atrial fibrillation; CAD: coronary artery disease; T2D: type 2 diabetes.

**Table 4 bioengineering-11-00872-t004:** Prediction performance of different diseases.

Disease	Model	Cor	AUC	Interval	Sen	Spec	Accuracy
CKD	PRS	−0.06	0.53	0.43~0.63	0.42	0.73	0.42
	IDP	0.34	0.68	0.59~0.76	0.87	0.47	0.66
	PRS + IDP	0.31	0.67	0.58~0.76	0.66	0.66	0.66
Asthma	PRS	0.16	0.59	0.54~0.63	0.40	0.74	0.56
	IDP	0.34	0.70	0.66~0.74	0.48	0.84	0.66
	PRS + IDP	0.40	0.73 *	0.69~0.77	0.61	0.73	0.67
COPD	PRS	0.54	0.82	0.73~0.92	0.84	0.71	0.78
	IDP	0.38	0.70	0.58~0.82	0.61	0.79	0.70
	PRS + IDP	0.66	0.88	0.81~0.96	0.79	0.89	0.83
AF	PRS	0.40	0.74	0.66~0.82	0.58	0.82	0.70
	IDP	0.37	0.71	0.62~0.79	0.76	0.58	0.68
	PRS + IDP	0.43	0.75	0.66~0.83	0.46	0.95	0.70
CAD	PRS	0.27	0.66	0.60~0.72	0.56	0.72	0.65
	IDPs	0.44	0.76	0.71~0.81	0.88	0.56	0.71
	PRS + IDP	0.50	0.81*	0.75~0.86	0.80	0.73	0.76
HF	PRS	0.22	0.63	0.47~0.79	0.74	0.55	0.66
	IDP	0.32	0.68	0.53~0.83	0.50	0.84	0.66
	PRS + IDP	0.50	0.79	0.66~0.91	0.73	0.81	0.77
Hypertension	PRS	0.12	0.57	0.53~0.61	0.50	0.61	0.56
	IDP	0.45	0.77	0.74~0.80	0.71	0.72	0.71
	PRS + IDP	0.40	0.73	0.70~0.76	0.65	0.71	0.68
MI	PRS	0.24	0.62	0.52~0.71	0.71	0.49	0.60
	IDP	0.29	0.68	0.58~0.77	0.56	0.75	0.66
	PRS + IDP	0.48	0.79	0.71~0.86	0.87	0.55	0.72
T2D	PRS	0.27	0.65	0.57~0.73	0.63	0.60	0.62
	IDP	0.57	0.84	0.78~0.90	0.71	0.86	0.78
	PRS + IDP	0.60	0.87 *	0.81~0.92	0.82	0.79	0.81

Cor: Pearson correlation; Sen: sensitivity; Spec: specificity; CKD: chronic kidney disease; COPD: chronic obstructive pulmonary disease; AF: atrial fibrillation; CAD: coronary artery disease; HF: heart failure; MI: myocardial infarction; T2D: type 2 diabetes, “*” indicates that PRSs + IDPs are significantly better than PRSs or IDPs alone.

**Table 5 bioengineering-11-00872-t005:** Prediction performance of different diseases with and without lifestyle factors.

Disease	Model	Cor	AUC	Interval	Sen	Spec	Accuracy
CKD	Baseline	0.31	0.67	0.58~0.76	**0.66**	0.66	0.66
	Baseline + drink	0.36	0.72	0.62~0.82	0.55	**0.82**	0.67
	Baseline + smoke	**0.37**	**0.72**	0.62~0.82	0.62	0.76	**0.69**
	Baseline + drink + smoke	0.31	0.68	0.58~0.78	0.49	0.82	0.65
Asthma	Baseline	0.35	0.70	0.66~0.74	0.59	**0.70**	0.65
	Baseline + drink	0.33	0.69	0.65~0.74	**0.78**	0.55	0.67
	Baseline + smoke	0.36	0.71	0.66~0.75	0.70	0.61	0.65
	Baseline + drink + smoke	**0.38**	**0.72**	0.67~0.76	0.65	0.70	**0.67**
COPD	Baseline	**0.71**	**0.91**	0.84~0.97	**0.92**	0.74	0.83
	Baseline + drink	0.55	0.84	0.72~0.97	0.79	**0.93**	**0.86**
	Baseline + smoke	0.42	0.76	0.62~0.89	0.71	0.75	0.73
	Baseline + drink + smoke	0.60	0.85	0.73~0.98	0.88	0.85	0.86
AF	Baseline	**0.50**	**0.81**	0.74~0.89	**0.76**	0.76	0.76
	Baseline + drink	0.39	0.78	0.68~0.87	0.61	**0.92**	0.76
	Baseline + smoke	0.49	0.79	0.70~0.88	0.64	0.88	0.74
	Baseline + drink + smoke	0.39	0.75	0.65~0.85	0.68	0.82	**0.78**
CAD	Baseline	0.46	0.77	0.72~0.82	0.66	0.79	0.73
	Baseline + drink	**0.51**	**0.80**	0.75~0.86	0.69	**0.84**	**0.76**
	Baseline + smoke	0.44	0.76	0.70~0.83	0.68	0.74	0.71
	Baseline + drink + smoke	0.40	0.73	0.66~0.79	**0.69**	0.69	0.69
HF	Baseline	**0.63**	**0.87**	0.78~0.97	0.85	0.81	**0.83**
	Baseline + drink	0.56	0.82	0.69~0.95	**0.90**	0.65	0.78
	Baseline + smoke	0.37	0.68	0.51~0.86	0.42	**1.00**	0.73
	Baseline + drink + smoke	0.24	0.58	0.40~0.77	0.64	0.56	0.60
Hypertension	Baseline	**0.47**	**0.78**	0.75~0.81	0.67	**0.76**	**0.72**
	Baseline + drink	0.46	0.77	0.74~0.81	0.77	0.65	0.71
	Baseline + smoke	0.45	0.77	0.73~0.80	**0.86**	0.56	0.71
	Baseline + drink + smoke	0.44	0.76	0.73~0.79	0.67	0.71	0.69
MI	Baseline	0.49	**0.80**	0.73~0.88	**0.87**	0.63	0.75
	Baseline + drink	0.48	0.78	0.69~0.87	0.83	0.71	**0.77**
	Baseline + smoke	**0.50**	0.79	0.70~0.88	0.79	0.66	0.73
	Baseline + drink + smoke	0.36	0.73	0.63~0.82	0.64	**0.75**	0.70
T2D	Baseline	**0.56**	**0.84**	0.78~0.90	0.67	**0.86**	0.76
	Baseline + drink	0.53	0.81	0.74~0.89	0.85	0.72	**0.78**
	Baseline + smoke	0.50	0.79	0.71~0.87	**0.90**	0.62	0.76
	Baseline + drink + smoke	0.52	0.80	0.73~0.88	0.85	0.63	0.73

Baseline means the model integrating PRSs and IDPs. Cor: Pearson correlation; Sen: sensitivity; Spec: specificity; CKD: chronic kidney disease; COPD: chronic obstructive pulmonary disease; AF: atrial fibrillation; CAD: coronary artery disease; HF: heart failure; MI: myocardial infarction; T2D: type 2 diabetes. Bolded text indicates the best performance for the respective metric.

## Data Availability

The raw data are available from UK Biobank via a standard application procedure at http://www.ukbiobank.ac.uk/register-apply accessed on 27 August 2023.

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
