# Peer review of "Integrating Multi-Organ Imaging-Derived Phenotypes and Genomic Information for Predicting the Occurrence of Common Diseases"

_bioengineering, 2024, doi:10.3390/bioengineering11090872_

Round 1

Reviewer 1 Report

Comments and Suggestions for Authors

In this study, Meng et al. focused on the fusion of imaging-derived phenotypes (IDPs) and polygenic risk scores (PRS) from different organs, including the brain, heart, lung, liver, spleen, pancreas, and kidney, for predicting the occurrence of nine common diseases. These diseases include atrial fibrillation, heart failure (HF), hypertension, myocardial infarction, asthma, type 2 diabetes, chronic kidney disease, coronary artery disease (CAD), and chronic obstructive pulmonary disease (COPD) in 8,646 individuals with complete IDPs and PRS data from approximately 500,000 participants in the UK Biobank. For each disease, three prediction models were developed utilizing imaging features, genomic data, and a fusion of both, and their performances were compared. Their results indicated that for seven diseases, the model integrating both imaging and genomic data achieved superior predictive performance. This is an interesting attempt to apply IDPs and PRS data for common disease prediction. However, I have several concerns:

Major Issues:

While genetic contributions are important in the pathogenesis of common diseases, they only represent limited heritability. Environmental exposures, such as smoking and alcohol use are also critical. The current study did not consider these factors in their predictive model and only included PRS with IDPs.

Given that this manuscript pertains to medical research with potential clinical applications, the prediction model should be validated in an external independent dataset before acceptance. The current manuscript only provides internal validation.

Minor Issues:

There are no tables presenting the distribution of the 8,646 individuals in terms of how many are healthy versus diagnosed with the investigated diseases. It is unclear whether patients with multiple diseases were included or excluded, and whether patients could be diagnosed with more than one disease simultaneously.

Comments on the Quality of English Language

Extensive editing of English language required

Reviewer 2 Report

Comments and Suggestions for Authors

The theme of the paper is interesting and important. The paper is well strucrured and written, hower it ha ssome weaknesses

- the methodology should be desribed in the manner to be repclicable

- the validation and verification of the  odels should be described in more detail

- the disscusion should be extended with comaprison to similar studies

Reviewer 3 Report

Comments and Suggestions for Authors

Thank you for the opportunity to revise this work.

The authors have created a system that combines phenotype data derived from imaging and genomic data to predict the occurrence of common diseases. Especially the focus is on: nine diseases 1. atrial fibrillation, 2. heart failure, 3. hypertension, 4. myocardial infarction, 5. asthma, 6. type 2 diabetes, 7. chronic kidney disease, 8. coronary artery disease and 9. chronic obstructive pulmonary disease. The developed systems are based on an available database UK biobank and on logistic regression model with lasso regularization.

The manuscript is in general well written, there are just a few comments that may help to improve it

1.       In the abstract is mentioned “the model integrating both imaging and genomic data achieved superior predictive performance.” Please clarify, superior compared to what?

2.       When comparing the ROC AUC of two methods, it is preferable to add a p-value for their comparison to indicate also if there is statistically significant difference. There are statistical methods to compare two ROC AUC, based on area and the relevant standard error, see for example on-line tools such as here https://www.medcalc.org/calc/comparison_of_independentROCtest.php This p-value can be added both in the abstract and in the manuscript body in the table that presents the outcomes. Furthermore statistical comparison of the sensitivity and specificity of the different methods would be of interest to appear.

3.       The first time mentioned the UK Biobank, please add a relevant reference

4.       Agree with the authors’ approach to include participants of European descent, however please explain the reasons on first appearance if such information, I.e. when referring to figure 3.

5.       Line 111: “Those IDPs were automatically extracted through 111 the deep learning model.”, Here is required more information for IDP extraction for all organs, propose to add one more column in table 2 and add references that describe the methodologies used to extract IDPs.

Comments on the Quality of English Language

Language is OK, minor issues detected

Round 2

Reviewer 2 Report

Comments and Suggestions for Authors

I have no further comments

Reviewer 3 Report

Comments and Suggestions for Authors

Dear colleagues

All review concerns are now addressed,

In my opinion this manuscript deserves to be published in Bioengineering.